# Sputnik Light and Sputnik V Vaccination Is Effective at Protecting Medical Personnel from COVID-19 during the Period of Delta Variant Dominance

**DOI:** 10.3390/vaccines10111804

**Published:** 2022-10-26

**Authors:** Gennady T. Sukhikh, Tatiana V. Priputnevich, Darya A. Ogarkova, Andrei A. Pochtovyi, Daria D. Kustova, Vladimir I. Zlobin, Denis Y. Logunov, Vladimir A. Gushchin, Alexander L. Gintsburg

**Affiliations:** 1Kulakov National Medical Research Center for Obstetrics, Gynecology and Perinatology of the Ministry of Health, 117997 Moscow, Russia; 2Federal State Budget Institution “National Research Centre for Epidemiology and Microbiology Named after Honorary Academician N F Gamaleya” of the Ministry of Health of the Russian Federation, 123098 Moscow, Russia; 3Department of Virology, Biological Faculty, Lomonosov Moscow State University, 119991 Moscow, Russia; 4Department of Infectiology and Virology, Federal State Autonomous Educational Institution of Higher Education I.M. Sechenov, First Moscow State Medical University of the Ministry of Health of the Russian Federation (Sechenov University), 119435 Moscow, Russia

**Keywords:** COVID-19, SARS-CoV-2, vaccine, Sputnik V, Sputnik Light

## Abstract

Medical personnel are a group of people that often encounter infectious agents, leading to greater risk of contracting infectious diseases. Specific prevention of diseases in this group is a priority. The epidemiological effectiveness of COVID-19 prevention in the group of medical workers due to the emergence of new variants of concern of the SARS-CoV-2 virus has not been studied in sufficient depth. We conducted a study of the effectiveness of vaccine use to protect medical workers at a large medical center for obstetrics and gynecology in Moscow. Sputnik V and Sputnik Light were the main vaccines used for the prevention of COVID-19. The vaccines are based on a variant of the S-protein of the SARS-CoV-2 virus, with adenovirus serotypes 5 and 26 as the vector for delivery. Vaccination of employees occurred during the period in which the Delta variant was spreading. The overall epidemiological effectiveness was 81.7% (73.1–87.6%) during the period in which the Delta variant was dominant. During the period from the beginning of vaccination (26 November 2020) until 8 February 2022, the overall effectiveness was 89.1% (86.9–91.0%). As expected, the highest effectiveness during this period was obtained in the group that received the third and fourth doses—96.5% (75.0–99.5%). The severity of COVID-19 in the vaccinated group was significantly lower than in the unvaccinated group.

## 1. Introduction

The disease COVID-19, caused by the SARS-CoV-2 virus, is extremely contagious and spreads mainly by airborne droplets [1,2]. Employees of medical organizations are at particular risk of infection due to their coming into contact with a large number of people [3,4,5]. In this regard, it is important to monitor the effectiveness of protection resulting from vaccination, specifically among medical personnel.

The main vaccines for the prevention of COVID-19 used in the Russian Federation are Sputnik V [6] and Sputnik Light [7]. These vaccines are based on the gene encoding the S-protein of the SARS-CoV-2 virus and adenoviruses as vectors for the delivery of the target antigen. Both vaccines have passed clinical trials and have been registered for use in COVID-19 prevention. According to a study by Barchuk et al., vaccination showed 81% protection against referral to hospital in patients with symptomatic SARS-CoV-2 infection during the third wave of the COVID-19 pandemic caused by the Delta VOC in St. Petersburg. During the study period, in St. Petersburg, among 1,227,496 individuals who received at least one dose of the vaccine, 96% received Sputnik V [8]. In the case of Sputnik Light vaccination, the epidemiological effectiveness in the group below 59 years of age was 75.28% (69.24–80.13%), whereas among older people, it decreased to 51.98% (35.61–64.19%) [9]. In the group of people living with HIV, the effectiveness of the Sputnik V vaccine was dependent on the level of CD4+ cells [10]. Thus, in the group with CD4+ < 350 cells/µL and CD4+ ≥ 350 cells/µL, the overall epidemiological effectiveness was 73.15% (50.27–85.50%) and 79.42% (72.54–84.57%), respectively. During the period of Delta variant dominance, even in the CD4+ ≥ 350 cells/µL group, the epidemiological effectiveness decreased to 65.34% (52.61–74.66%). At the same time, protection from hospitalization and severe course in this group remained at levels of 75.77% (44.25–89.47%) and 93.05% (49.51–99.04%), respectively.

The effectiveness of the use of Sputnik V and Sputnik Light among medical personnel has not been investigated. In order to evaluate this, we organized a cohort study of the epidemiological effectiveness of the vaccine for the prevention of COVID-19 among employees of the Kulakov National Medical Research Center for Obstetrics, Gynecology and Perinatology of the Ministry of Health.

## 2. Materials and Methods

### 2.1. Database Preparation for Epidemiological Analysis

We systematically collected and analyzed data on 2621 medical workers employed at the Kulakov National Medical Research Center for Obstetrics, Gynecology and Perinatology of the Ministry of Health. Employees with an unknown date of vaccination or date of the first disease were removed from the database. A total of 914 individuals were vaccinated with vaccines other than Sputnik V or Sputnik Light, and were also not taken into account. For 29 people, the break between the first and second components of their Sputnik V vaccination was shorter than 14 days. When analyzing the vaccine’s effectiveness, we also excluded records with cases before the observation period or where vaccination took place less than 14 days before the end of observation. Individuals for whom the period between vaccination and disease was shorter than 14 days were also not taken into account. Database preparation process is shown in the Figure 1. 

We studied vaccine effectiveness for two time periods: (i) Delta variant dominance period; and (ii) the period from vaccination commencement (26 November 2020) to data upload on 8 February 2022. We analyzed 1287 records throughout the entire period from 26 November 2020 to 8 February 2022, and 1092 records for the Delta dominance period.

Written informed consent was obtained from all subjects in accordance with the order of the Ministry of Health of the Russian Federation of 21 July 2015 #474 n. This study was reviewed and deemed exempt by the Local Ethics Committee of the Gamaleya Center (protocol No. 14, 29 September 2021).

### 2.2. Statistical Analysis

Statistical analysis was performed using the IBM SPSS Statistics program, version 26 and the R programming environment. The evaluation of statistically significant differences in qualitative characteristics was performed using Pearson’s chi-squared criterion. The differences were considered statistically significant at *p* < 0.05. In pairwise comparisons of multi-field tables, the Benjamin–Hochberg adjustment for multiplicity was used.

To compare the age in different groups, the Student’s T-test was used for paired comparisons and a one-factor ANOVA for comparing more than two groups.

The effectiveness of the vaccine was calculated using Formula (1):(1 − RR) × 100%(1)
where RR is the risk ratio, as well as using Formula (2):(1 − HR) × 100%(2)
where HR is the hazard ratio, calculated using Cox regression.

Confidence intervals were calculated using the Taylor series [11] and Cox regression.

### 2.3. Characteristics of the Genetic Landscape of SARS-CoV-2 Lines

To characterize the landscape of the SARS-CoV-2 circulating lines during the observation period, data from the GISAID database were used, obtained using the following queries: “Location: Europe/Russia/Moscow”, “Host: Human”, “Complete sequence:Yes”, “Collection date complete: Yes” and “Sequence length” ≥ 20,000 (Request date 12 September 2022). Information about the genetic lines of the remaining 7306 sequences was grouped as follows: Wuhan (B.1 + B.1.x), Alpha (B.1.1.7 + Q.x), Beta (B.1.351 + B.1.351.x), Delta (B.1.617.x + AY.x), Omicron (B.1.1.529 + BA.x). The remaining lines were combined into the “Other” group. The obtained results were combined with metadata with subsequent plotting in the R environment using dplyr packages [12] and ggplot2 [13].

## 3. Results

### 3.1. Circulating Lines SARS-CoV-2 Virus

Since the beginning of the pandemic in Moscow, there have been five waves of increased incidence of COVID-19. This was primarily due to changes in the composition of circulating SARS-CoV-2 genetic lines and the appearance of additional mutations in the spike protein receptor binding domain [14]. The dominant lines from the beginning of 2020 to December 2020 were B.1 and its subvariants (B.1.x), which accounted for almost 100% of the total diversity (Figure 2A). At the end of December 2020, the first cases of infection with the Alpha variant were noted, which led to the partial displacement of B.1 + B.1.x lines, reducing their share to approximately 53%. In February–March 2021, a rapid increase in the Delta variant occurred, which displaced all other circulating lines until November 2021. A similar pattern was observed for Omicron, which has been the dominant line since January 2022.

### 3.2. Vaccination and Cases of the Disease among the Medical Center Personnel

We conducted observations on medical workers at the Kulakov National Medical Research Center for Obstetrics, Gynecology and Perinatology of the Ministry of Health in Moscow during 25 months of the pandemic. The dynamics of the incidence of COVID-19 at the medical center, in general, reflect the nature of the incidence in Moscow [15,16] (Figure 2A,B). There are several peaks of morbidity: (i) April–May 2020 and (ii) October 2020–January 2021, which were caused mainly by the circulation of the original strains B.1 and B.1.x; (iii) June–August 2021 and (iv) October–December 2021, which were mainly due to the Delta strain; and (v) January–February 2022, which was due to the Omicron strain (Figure 2A).

Vaccination of the center’s employees began in October 2020; however, the peak of vaccination occurred in June–August 2021 (Figure 2C). This trend also reflects the dynamics of mass vaccination that was taking place in Moscow [17]. In entire cohort of 1624 people was included, 1609 (99.1%) of whom were vaccinated at least once during the entire observation period. A total of 805 people (49.6%) experiences no cases of COVID-19 during the observation period, 704 (43.3%) people contracted COVID-19 once, 113 (7.0%) contracted it twice, and two (0.1%) contracted three cases of confirmed RT-PCR COVID-19 disease during the observation time.

Most people who received two or more doses of the vaccine did so according to the following scheme: Sputnik V for vaccination and Sputnik Light for booster immunization. Analysis of the time elapsed between the receipt of doses shows that the average time between the first and second doses was 23.0 days (95% CI: 22.3–23.7 days). At the same time, only 54 people out of the 951 (5.7%) considered in the group to have received two or more doses had a time between the first and second dose of more than 30 days. In addition, the time between the first and second dose was more than 60 days in only six (0.63%). These deviations could be associated with employees’ vacations or medical reasons. Almost all of the observed volunteers, without taking into account the disease and the observation period, received the third vaccination 140–303 days after the second (average: 208 days, 95% CI: 199–217 days). In addition, only one person had only 21 days between the second and third stages of vaccination. During revaccination, only three people among those considered in the “three or four doses” group received four injections of the vaccine. Thus, it can be assumed that our study relates mainly to people who took Sputnik V as part of their vaccination and Sputnik Lite as a booster injection.

### 3.3. Effectiveness of Sputnik V in the Medical Personnel

To calculate the effectiveness of vaccination with Sputnik V, the frequency of COVID-19 disease in people with and without vaccination was analyzed. The analysis included 1624 employees of a medical research center with known dates of disease who were not vaccinated, or were vaccinated with Sputnik V or Sputnik Light. We excluded people who became ill less than 14 days after vaccination. Diseases occurring before 26 November 2020 (the date of vaccination commencement) were excluded from the analysis. Additionally, those who became ill less than 14 days after vaccination were excluded from the study, as well as those who were vaccinated less than 14 days before the end of observation (8 February 2022). In this overall vaccine effectiveness analysis, we used data for about 1287 medical workers.

As shown in Figure 2A, the dominance of the Delta variant lasted from May to December 2021, but the Omicron strain had already appeared in December. That could influence the effectiveness of the vaccine, so we prepared a second vaccine effectiveness analysis for the Delta dominance period only, before the appearance of Omicron. To do this, we excluded cases before 1 May 2021 as well as people vaccinated less than 14 days before 30 November 2021. Therefore, we had 1092 records in the database to calculate vaccine effectiveness during the dominance of the Delta variant of the SARS-CoV-2 virus.

The analysis compared the frequency of disease in the unvaccinated with the frequency of disease after vaccination.

To calculate the effectiveness, Formula (1) (see Materials and Methods) was used, and confidence intervals were calculated using the Taylor series [11]. Taking into account the importance of observation time for each person, we also calculated the epidemiological effectiveness of vaccination using Cox regression with Formula (2).

The effectiveness over the entire follow-up period is shown in Appendix A, and the effectiveness over the Delta dominance period is shown in Table 1.

The effectiveness, calculated using RR, presented some contradictions. For example, vaccine effectiveness for the entire period was 73.7% (95% CI: 70.9–76.2) (Appendix A), and for the Delta dominance period, VE = 83.2% (95% CI: 77.8–87.3%) (Table 1). This unexpected increase in vaccine effectiveness could be due to the shorter period of Delta dominance compared to the entire observation period. A similar increase in effectiveness could be seen for those who had received only one does of Sputnik V compared to VE after the second dose. The average time between receiving the first and second doses was 23.0 ± 11.8 days with 95% CI: 22.3–23.7 days, so there was a very short period of time in which to follow up people who had received only one dose.

Thus, VE calculated using Cox regression more accurately represents the data, since it uses observation time as one of the arguments. Therefore, overall VE without taking into account the number of vaccinations was 89.1% (95% CI: 86.9–91.0%) for the entire period and 81.7% (95% CI: 73.1–87.5%) for the Delta dominance period. The cumulative risk graph is shown in Figure 3. The greatest epidemiological effectiveness was obtained in people who had received their third or fourth vaccine doses, which was 96.5% (75.0–99.5%) for entire period. There was not a single case after receiving the third or fourth doses of Sputnik V during the Delta dominance period, therefore, the effectiveness presented in Table 1 may not reflect the reality.

Age was known for 1578 people (97.2% of the 1624 of the original database). The average age of all studied employees was 43.56 ± 13.09 years (95% CI: 42.92–44.21). For the population included in the analysis of the effectiveness of vaccination during the dominance of the Delta variant of the SARS-CoV-2 virus, data on age were available for 1048 people out of 1092 (96.0%). The average age, standard deviations and 95% confidence intervals are presented in Table 2, depending on the number of vaccine doses and the presence of a confirmed case of COVID-19. No significant differences were found in any of the studied groups, the age was homogeneous, and amounted to 43.30 ± 13.36 years (95% CI: 42.49–44.11). Demographic data were available for 1048 people out of 1092 (96.0%) for the population included in the vaccine effectiveness analysis during the dominance of the Delta variant. The average age, standard deviations and 95% confidence intervals are presented in Table 2, depending on the number of vaccine doses and the presence of a confirmed case of COVID-19. No significant differences were found in any of the studied groups, and the age was homogeneous and amounted to 43.30 ± 13.36 years (95% CI: 42.49–44.11).

To analyze the effect of vaccination on the disease severity, all cases of the disease for which the severity was known were taken, except for cases where disease began less than 14 days after the last administered dose of Sputnik V (*n* = 799). The severity was classified using WHO recommendations [18]. The severity of the disease differed in patients with and without vaccination (*p* = 0.008, chi-squared criterion) (Table 3). The proportion of patients with mild course was higher in the vaccinated group (in the people who received at least one dose) (*p* = 0.002, Benjamin–Hochberg adjustment). The proportion of cases with mild course was 77.5% (*n* = 221) in the vaccinated group and 67.3% (*n* = 346) in the unvaccinated group. The proportion of moderate patients was higher in the unvaccinated group, at 29.4% (*n* = 151), where in the vaccinated group this proportion was 20.7% (*n* = 59). The differences were significant (*p* = 0.008, Benjamin–Hochberg adjustment). The proportions of patients with severe disease were 1.8% (*n* = 5) and 3.3% (*n* = 17) in the vaccinated and unvaccinated groups, respectively. No significant differences were found in the proportions of the severe course of the disease, which might be due to an insufficient number of severe cases (*n* = 23) (*p* = 0.199, Benjamin–Hochberg adjustment). Additionally, we found no significant differences between average age in groups with different vaccination status and case severity (Appendix A), but a tendency of increased average age in groups with more severe courses of the disease was observed for both unvaccinated and vaccinated medical workers. For unvaccinated people, the average ages were 43.37 (42.08–44.66), 44.87 (42.99–46.75) and 49.47 (42.07–56.88) years old for mild, moderate and severe courses of COVID-19, respectively. In addition, for vaccinated persons, the average ages were 41.32 (39.66–42.97), 44.81 (41.22–48.41) and 48.20 (40.58–55.82) years old for the same respective degrees of case severity.

## 4. Discussion

Mass vaccination with the drugs for the prevention of COVID-19 could not prevent the further spread of the SARS-CoV-2 virus. This is partly due to the slow pace of the introduction of the vaccine around the world, including the rejection of vaccines by some groups in society [19,20]. An objective problem is also the emergence and spread of new variants of the virus that can avoid the immune response [21]. In this regard, the study of the vaccine effectiveness in risk groups becomes especially relevant.

The effectiveness of Sputnik V and Sputnik Light was studied in a group of elderly people, as well as a group of people living with HIV [9,10]. Healthcare workers are also at risk for COVID-19 [22,23]. Medical workers provide assistance according to the profile of activity, regardless of the presence of an infectious disease. A significant risk has been shown for dentists, with job leaving intentions increasing among specialists in this industry [24]. Specific prevention of COVID-19 among medical personnel is an important component of the protection of this group. To assess the effectiveness of the use of prevention with Sputnik V and Sputnik Light, we conducted a retrospective cohort study on the basis of the Kulakov National Medical Research Center for Obstetrics, Gynecology and Perinatology of the Ministry of Health.

During the 25 months of the pandemic, the initial variants of the virus [14] and the Delta variant [25] circulated. The incidence of COVID-19 in the medical center under study reflected the nature of the incidence in Moscow [15]. The peak of the vaccination of medical workers occurred in June–August 2021, which also reflected the dynamics of mass vaccination taking place in Moscow [17]. A total of 1624 people were studied, almost all of whom were vaccinated by the end of the observation period (99.1%). A total of 805 people (49.6%) had no history of COVID-19, 704 people (43.3%) had COVID-19 once, 113 (7.0%) had it twice, and two (0.1%) had three cases of confirmed PCR COVID-19 diseases. The Cox regression model proved to be more applicable for calculating the epidemiological vaccine effectiveness in a cohort of medical workers. The effectiveness values were 89.1% (86.9–91.0%) and 81.7% (73.1–81.5%) for the entire period and for the Delta dominance period, respectively. The epidemiological efficacy for the entire period increased to 96.5% (75.0–99.5%) among those who had received three or four doses of the vaccine. Increasing protection against COVID-19 when using booster immunizations is a well-known phenomenon [26]. In the case of the Sputnik V and Sputnik Light drugs, the clinical effectiveness of booster doses was also shown for the Omicron variant [27], which differs significantly from previously circulating variants [28]. A marker IgGs level of 142.7 BAU/mL was recommended, a decrease in which indicates the necessity of booster immunization [29].

The analysis of case severity shows that employees at the medical research center who became ill following vaccination had lower disease severity. The proportion of cases with mild course was 77.5% (*n* = 221) in the vaccinated group and 67.3% (*n* = 346) in the unvaccinated group, while the proportion of patients with a moderate course was higher in the unvaccinated group at 29.4% (*n* = 151), than it was the vaccinated group, at 20.7% (*n* = 59). These differences were statistically significant (*p* < 0.05, Benjamin–Hochberg adjustment).

The CDC website provides information about four programs for evaluating the effectiveness of vaccines against COVID-19 among medical workers [30]. Each of the programs includes several research centers: Arctic Investigations Program (AIP) (two sites), Emerging Infections Program (EIP) (10 sites), Preventing Emerging Infections through Vaccine Efficacy Testing (the PREVENT project) (16 sites), and Safety and Healthcare Epidemiology Prevention Research Development (the SHEPheRD program) (six sites). Results have been partially published; in particular, PREVENT has evaluated the efficacy of the Pfizer-BioNTech and Moderna products. One study showed that the VE of a single dose (measured 14 days after the first dose through 6 days after the second dose) was 82% (95% CI: 74–87%), whereas the adjusted VE of two doses (measured ≥7 days after the second dose) was 94% (95% CI: 87–97%) [31]. In another study, Pfizer-BioNTech vaccine efficacy for partial vaccination was 77.6% (95% CI: 70.9 to 82.7) with the BNT162b2 vaccine and 88.9% (95% CI: 78.7 to 94.2) with the mRNA-1273 vaccine (Moderna); for complete vaccination, vaccine effectiveness was 88.8% (95% CI: 84.6 to 91.8) and 96.3% (95% CI: 91.3 to 98.4), respectively [32]. However, both studies were conducted in the period before the circulation of the Delta variant. Data on the effectiveness of the vaccines during the period of the dominance of the Delta variant have not yet been published. Our study expands the body of data on the effectiveness of vaccines among medical personnel under the conditions of the arrival of new virus variants and taking into account the use of booster doses. The results obtained indicate that among the medical workers who received booster doses, the effectiveness of the protection offered by the Sputnik V vaccine remained the greatest.

Our research has some limitations. For example, at present, the most epidemiologically significant variant is the Omicron and its sublines. Our cohort study does not allow us to calculate the epidemiological vaccine effectiveness in relation to the Omicron variant, since by the time it appeared in our cohort there were no unvaccinated patients left. We will continue the study of this cohort using a different design that takes into account not only the fact of vaccination, but also the time that has passed since vaccination, as well as the degree of immunity tension using methods for assessing humoral and cellular immunity.

## 5. Conclusions

This study showed that the vaccination of medical workers as a risk group for COVID-19 with Sputnik V and Sputnik Light before the appearance of the Omicron variant reliably protected against the disease, as well as reducing the severity of the clinical course. Booster immunizations with three and four doses increased the effectiveness of protection.

## Figures and Tables

**Figure 1 vaccines-10-01804-f001:**
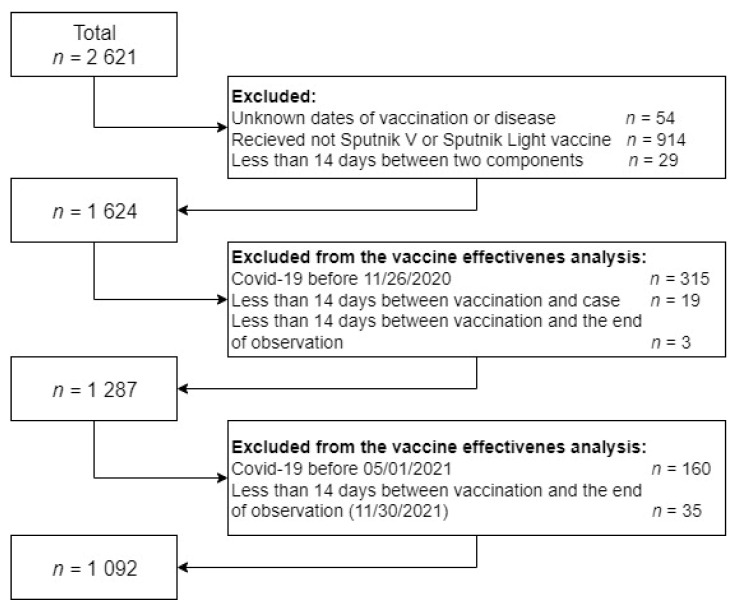
Database in preparation for the analysis.

**Figure 2 vaccines-10-01804-f002:**
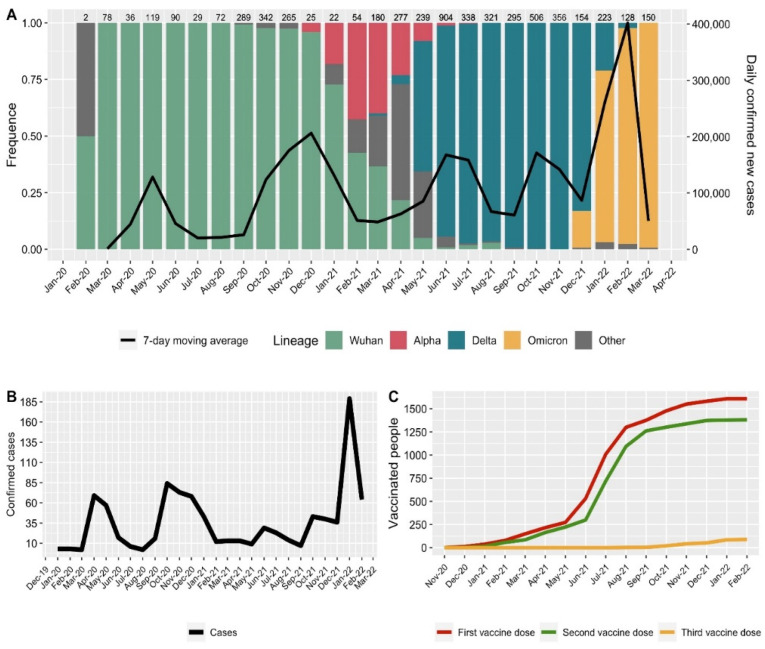
(**A**) Distribution of SARS-CoV-2 genetic lines in Moscow from the beginning of the pandemic until March 2022, according to GISAID. The genetic lines are grouped according to lineage. The “Other” group consists of the least represented lines. The numerical value above the bar plot indicates the number of valid sequences in the GISAID database for a specific period. The black line shows the 7-day moving average number of daily COVID-19 cases in Moscow according to [15]. (**B**) Cases in the medical research center. (**C**) Dynamics of the number of vaccinated medical center personnel. Color indicates the number of doses.

**Figure 3 vaccines-10-01804-f003:**
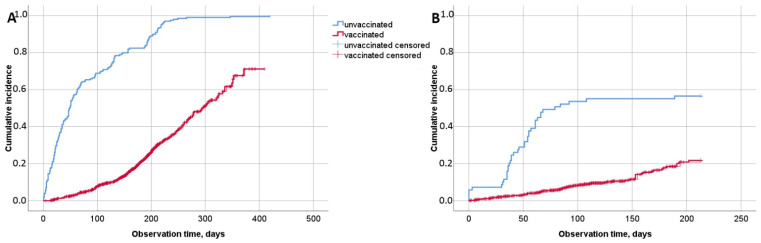
The cumulative risk of medical center employees, depending on the availability of vaccination. (**A**) For the entire period, HR = 0.109 (95% CI: 0.090–0.131), which corresponds to VE = 89.1% (95% CI: 86.9–91.0%); (**B**) for the Delta dominance period, HR = 0.167 (95% CI: 0.113–0.245), which corresponds to VE = 83.3% (95% CI: 75.5–88.7%).

**Table 1 vaccines-10-01804-t001:** Cases distribution of the medical center’s employers, depending on the vaccination status during Delta dominance period (1 May–30 November 2021).

Vaccination Status	Cases	Non-Cases	VE, % (95% CI)	
			(1-RR)	(1-HR)
Unvaccinated	39	30		
One dose	6	113	91.1% (80.0–96.0%)	78.8% (49.2–91.1%)
Two doses	91	788	81.7% (75.7–86.7%)	81.8% (73.1–87.6%)
Three or four doses	0	25	100% (–)	100% (–)
At least one dose	97	926	83.2% (77.8–87.3%)	81.7% (73.1–87.5%)

**Table 2 vaccines-10-01804-t002:** Average age depending on vaccination status and COVID-19 history for the Delta variant of SARS-CoV-2 virus dominance period.

Vaccination Status	Average Age, Years Old M ± SD (95% CI)	With COVID-19 during the Observation Period	Without COVID-19 during the Observation Period	*p* (Student’s Criterion)
Unvaccinated (*n* = 64, 92.8%)	42.98 ± 12.77 (39.79–46.17)	*n* = 39 (100%) 44.00 ± 12.81 (39.85–48.15)	*n* = 25 (83.3%) 41.40 ± 12.80 (36.12–46.68)	**0.431**
One dose (*n* = 105, 88.2%)	42.28 ± 13.73 (42.62–47.93)	*n* = 6 (100%) 44.17 ± 10.83 (32.80–55.54)	*n* = 99 (87.6%) 45.34 ± 13.93 (42.57–48.12)	**0.840**
Two doses (*n* = 854, 97.2%)	42.97 ± 13.31 (42.97–42.07)	*n* = 91 (100%) 44.14 ± 13.16 (41.40–46.88)	*n* = 763 (96.8%) 42.83 ± 13.33 (41.88–43.77)	**0.373**
Three or four doses 25 (100%)	47.00 ± 14.35 (41.08–52.92)	–	*n* = 25 (100%) 47.00 ± 14.35 (41.08–52.92)	**–**
** *p (ANOVA)* **	**0.189**	**0.998**	**0.135**	

**Table 3 vaccines-10-01804-t003:** Severity of case depending on vaccination status.

Severity	Non-Vaccinated (*n* = 514)	Vaccinated (*n* = 285)
Mild	346 (67.3%)	221 (77.5%)
Moderate	151 (29.4%)	59 (20.7%)
Severe	17 (3.3%)	5 (1.8%)

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
