# Peer review of "Sputnik Light and Sputnik V Vaccination Is Effective at Protecting Medical Personnel from COVID-19 during the Period of Delta Variant Dominance"

_vaccines, 2022, doi:10.3390/vaccines10111804_

Round 1

Reviewer 1 Report

In their manuscript Sukhikh et al. provide data on the efficacy of Sputnik Light and Sputnik V vaccines in protecting medical personal against SARS-CoV-2 infection and COVID-19 disease. In response to the COVID-19 pandemic several vaccines have been developed and evaluated among them mRNA-based vaccines of Pfizer/BioNTech or Moderna and the chimpanzee adenovirus vector-based vaccine of AstraZeneca. The two-component recombinant human adenovirus (rAd26 and rAd5) Sputnik V and the one component (rAd26) Sputnik Light belongs to the adenovirus-based vaccine vectors. Whereas vaccines of Pfizer/BioNTech, Moderna and AstraZeneca are intensively evaluated for their reactogenicity, immunogenicity and efficacy in numerous clinical studies, there are less data available on Sputnik V and Sputnik Light with this regard. As COVID-19 pandemic is still a major challenge for the health system especially in the view of new emerging variants, there is an urgent need for evaluation of further alternatives of already existing vaccines and drugs. Therefore, the present study provides important information on a less investigated vaccine.

However, there are several weaknesses:

Generally, the authors investigate groups with one, two doses as well as three or four (booster) doses of vaccination. In this regard it is not clear whether two doses group also includes individuals with one dose Sputnik Light followed with a second booster later on. Therefore, VE should be provided separately in groups after one dose Sputnik Light, two doses Sputnik V as well as Sputnik Light (one shot) with one/two booster and Sputnik V (two shots) with one/two booster. Furthermore, in Table 1. is misleading that the overall VE is in the same row with the non-vaccinated group. Please provide a row with the number of cases and non-cases from all vaccinees (independent on the number of vaccine shots), which were used for calculation of the total VE.

Demographic information (Table) for all study groups would be helpful to understand differences in the VE found especially for the one dose and two doses. In this regard, it is assumed that vaccinations with Sputnik Light started later then with Sputnik V. Might be this a reason for the difference in the VE (e.g. Sputnik Light vaccinees experienced less high incidence periods of infection…). Potential explanations found in VE between one dose and two doses should be extended, because this is a not expected outcome of the study.

Discussion largely repeats results instead of discuss data.   

Ethic statement is missing.

For case severity information should be provided how mild, moderate and severe were defined. Again, case severity for all study groups including demography (age, sex) would be helpful. 

The authors use terms “medicines or drugs” for adenovirus-based SARS-CoV-2 vaccine vectors throughout the text, which is not correct as a vaccine is rather a medical product that stimulates a person’s immune system to produce immunity to a specific disease, protecting the person from that disease. Please use the term “vaccine” instead.

Line 42 – “Wuhan virus variant” is the original ancestral strain of SARS-CoV-2, I would not name it as “variant”. 

Author Response

Thank you for valuable comments concerning the manuscript. We provided the additional information and made appropriate corrections along the text.

Response to Reviewer 1 Comments

In their manuscript Sukhikh et al. provide data on the efficacy of Sputnik Light and Sputnik V vaccines in protecting medical personal against SARS-CoV-2 infection and COVID-19 disease. In response to the COVID-19 pandemic several vaccines have been developed and evaluated among them mRNA-based vaccines of Pfizer/BioNTech or Moderna and the chimpanzee adenovirus vector-based vaccine of AstraZeneca. The two-component recombinant human adenovirus (rAd26 and rAd5) Sputnik V and the one component (rAd26) Sputnik Light belongs to the adenovirus-based vaccine vectors. Whereas vaccines of Pfizer/BioNTech, Moderna and AstraZeneca are intensively evaluated for their reactogenicity, immunogenicity and efficacy in numerous clinical studies, there are less data available on Sputnik V and Sputnik Light with this regard. As COVID-19 pandemic is still a major challenge for the health system especially in the view of new emerging variants, there is an urgent need for evaluation of further alternatives of already existing vaccines and drugs. Therefore, the present study provides important information on a less investigated vaccine.

However, there are several weaknesses:

Generally, the authors investigate groups with one, two doses as well as three or four (booster) doses of vaccination. In this regard it is not clear whether two doses group also includes individuals with one dose Sputnik Light followed with a second booster later on. Therefore, VE should be provided separately in groups after one dose Sputnik Light, two doses Sputnik V as well as Sputnik Light (one shot) with one/two booster and Sputnik V (two shots) with one/two booster.

Response: Thank you for your question. We have added this information to the text to section 3.2. Two doses group include people who got two doses only. Time between vaccinations was not taken into account and only the number of  received components was considered, since the absolute majority of people who took part in the study were vaccinated according to the basic scheme when two-component Sputnik V (1st dose rAd26 and 2nd dose rAd5) was used as primary immunization) and then Sputnik Light (third dose rAd26) or Sputnik V (3rd dose rAd26 and 4th dose rAd5)) during revaccination. Analysis of the time elapsed between receiving doses shows that the average time between the first and second component was 23.0 days (95% CI: 22.3-23.7 days). With the recommended Sputnik V vaccination regimen, an interval of 21 days is recommended between the first dose of rAd26 and the second dose of rAd5, so that vaccination was carried out practically without violation of deadlines. At the same time, only 54 people out of 951 (5.7%) (who were considered in the group who delivered two or more components) had a time between the first and second dose of more than 30 days. And only in 6 (0.63%) the time between the first and second dose was more than 60 days. Such deviations are usually associated with people's vacation or medical reasons so a person cannot deliver a second dose on time. Almost all the observed volunteers, without taking into account the disease and the observation period, received the third vaccination 140-303 days after the second (Average: 208 days, 95% CI: 199 – 217 days). And only one person had only 21 days between the second and third stages of vaccination. Thus, we can assume that our study relates mainly to people who have taken two doses of Sputnik V as part of vaccination and Sputnik Lite as a booster injection designed to increase immunity tension. During revaccination, only 3 people from those who were considered in the "three or four doses" group made 4 injections of the drug.

Furthermore, in Table 1. is misleading that the overall VE is in the same row with the non-vaccinated group. Please provide a row with the number of cases and non-cases from all vaccinees (independent on the number of vaccine shots), which were used for calculation of the total VE.

Response: We agree with your comment and have made changes to table 1. Also we changed data in this table according to other reviewer’s comment.

Demographic information (Table) for all study groups would be helpful to understand differences in the VE found especially for the one dose and two doses. In this regard, it is assumed that vaccinations with Sputnik Light started later then with Sputnik V. Might be this a reason for the difference in the VE (e.g. Sputnik Light vaccinees experienced less high incidence periods of infection…). Potential explanations found in VE between one dose and two doses should be extended, because this is a not expected outcome of the study.

Response: Thank you for your advice. We have added a table with demographic information in different groups during Delta dominance. No significant differences were found in any of the studied groups, the age was homo-geneous and amounted to 43.30±13.36 years (95% CI: 42.49 – 44.11).

Discussion largely repeats results instead of discuss data. 

We have added more discussion of our results in the discussion section now.

Ethic statement is missing.

Response: We have added information about ethic committee decision and about informed consent in materials and methods

For case severity information should be provided how mild, moderate and severe were defined. Again, case severity for all study groups including demography (age, sex) would be helpful. 

Response: Thanks for the note, the severity was determined according to the WHO (WHO. Clinical Management of Covid-19—Interim Guidance. Available online: https://apps.who.int/iris/bitstream/handle/10665/332196/WHO-2019-nCoV-clinical-2020.5-eng.pdf?sequence=1&isAllowed=y ). We added this information to the manuscript.

  We also added data on the age of the patients. This demographic data  was added in supplementary table 2.

The authors use terms “medicines or drugs” for adenovirus-based SARS-CoV-2 vaccine vectors throughout the text, which is not correct as a vaccine is rather a medical product that stimulates a person’s immune system to produce immunity to a specific disease, protecting the person from that disease. Please use the term “vaccine” instead.

Response: We agree with this remark, we have checked the text and corrected the style, corrected some terms and improved our English

Line 42 – “Wuhan virus variant” is the original ancestral strain of SARS-CoV-2, I would not name it as “variant”. 
Response: Thank you for your remark, we have replaced this in the text.

Reviewer 2 Report

COVID-19 vaccine effectiveness draws extensive attention from the government, research institutes, and the public. In this manuscript, Alexander L. Gintsburg and his co-workers investigated the effectiveness of Sputnik Light and Sputnik V vaccines in protecting medical workers from Delta variant infection. This study collected data from 2621 medical staff at a medical research center in Moscow and analyzed vaccine effectiveness during the Delta variant prevalence. The results showed that the overall vaccine effectiveness was 73.7%. After receiving three or four doses of the vaccine, the epidemiological effectiveness reached 97.5%. I believe this study provides valuable information to understand better the effectiveness of Sputnik Light and Sputnik V vaccines.

The benefits of this study are below.

1. The study investigated the vaccine's protective effect during the Delta variant's circulation. The authors drew a clear timeline of the SARS-CoV-2 variants' circulation in Moscow based on the reported sequences, giving sufficient temporal evidence that the time interval of the investigated samples indeed fell during the local Delta epidemic.

2. The background of the sample cohort was clear, and the sample size was enough to be statistically significant. Also, the statistical analysis used in this study is reasonable.

3. The presentation of the study is clear, and the figures and tables are well-organized, showing enough details and information.

On the other side, some points could be improved.

1. The authors analyzed data between 11/26/2020 (starting time of vaccination) and 01/25/2022 (end of observation). However, as shown in figure 2, Delta variant dominance in Mosco occurred from May to December 2021, and the medical workers' vaccination rate rapidly increased between June and August 2021. Therefore, the authors should consider narrowing the investigated time to May-December 2021. The survey results would give a more realistic picture of the vaccine's effectiveness against the Delta variant, as the Delta variant was not the dominant variant, and the vaccinated staff was less than 15% (250/1624) before May 2021, based on figure 2.

2. There are a few syntax mistakes. For example, the title missed a staff or workers after the word "medical."

3. Citation in the introduction should be accurate. For instance, in line 45, reference eight did not conclude: "the effectiveness of Sputnik V in the general population during the spread of the Delta variant averaged 80% ".

4. Please update references to the latest version. For example, reference eight has been published in BMC Medicine (PMID: 36123681).

Author Response

Thank you for valuable comments concerning the manuscript. We provided the additional information and made appropriate corrections along the text.

Response to Reviewer 2 Comments

COVID-19 vaccine effectiveness draws extensive attention from the government, research institutes, and the public. In this manuscript, Alexander L. Gintsburg and his co-workers investigated the effectiveness of Sputnik Light and Sputnik V vaccines in protecting medical workers from Delta variant infection. This study collected data from 2621 medical staff at a medical research center in Moscow and analyzed vaccine effectiveness during the Delta variant prevalence. The results showed that the overall vaccine effectiveness was 73.7%. After receiving three or four doses of the vaccine, the epidemiological effectiveness reached 97.5%. I believe this study provides valuable information to understand better the effectiveness of Sputnik Light and Sputnik V vaccines.

The benefits of this study are below.

  1. The study investigated the vaccine’s protective effect during the Delta variant’s circulation. The authors drew a clear timeline of the SARS-CoV-2 variants’ circulation in Moscow based on the reported sequences, giving sufficient temporal evidence that the time interval of the investigated samples indeed fell during the local Delta epidemic.
  2. The background of the sample cohort was clear, and the sample size was enough to be statistically significant. Also, the statistical analysis used in this study is reasonable.
  3. The presentation of the study is clear, and the figures and tables are well-organized, showing enough details and information.

On the other side, some points could be improved.

  1. The authors analyzed data between 11/26/2020 (starting time of vaccination) and 01/25/2022 (end of observation). However, as shown in figure 2, Delta variant dominance in Mosco occurred from May to December 2021, and the medical workers’ vaccination rate rapidly increased between June and August 2021. Therefore, the authors should consider narrowing the investigated time to May-December 2021. The survey results would give a more realistic picture of the vaccine’s effectiveness against the Delta variant, as the Delta variant was not the dominant variant, and the vaccinated staff was less than 15% (250/1624) before May 2021, based on figure 2.

Response: We agree with your comment and have added the analysis of vaccine effectiveness during Delta dominance period, but in December the Omicron strain have already appeared in Moscow. That could influence the vaccine effectiveness, so we have prepared the second vaccine effectiveness analysis only for the Delta dominance period before The Omicron appearance. So, we have analyzed the period from 05/01/2021 to 11/30/2021. Also, we have replaced Table 1 to VE during Delta dominance period and moved the original table to Supplementary.

  1. There are a few syntax mistakes. For example, the title missed a staff or workers after the word “medical.”

Response: We agree with this remark, we have checked the text and corrected the style, corrected some terms and improved the translation.

  1. Citation in the introduction should be accurate. For instance, in line 45, reference eight did not conclude: "the effectiveness of Sputnik V in the general population during the spread of the Delta variant averaged 80% ".

Response: Thank you for your comment, we have clarified the text. «According to study by Barchuk et al., vaccination showed 81% protection against referral to hospital in patients with symptomatic SARS­CoV­2 infection during the third wave of COVID­19 pandemic caused by Delta VOC in St. Petersburg. In study period in St. Petersburg, among 1,227,496 individuals who received at least one vaccine dose of vaccine 96% received Sputnik V [PMID: 36123681

  1. Please update references to the latest version. For example, reference eight has been published in BMC Medicine (PMID: 36123681).

Response: Thanks for the note, references have been corrected at the time of downloading the manuscript

Reviewer 3 Report

Title is missing the word, workers or staff.  The study examine the ability of Sputnik vaccines to protect medical staff from Covid-19.  Staff with 1,2 3 or four doses were surveyed with benefits found at all dose levels, although the small number for 3 to 4 doses is perhaps too small for any real conclusion.

Table 1 was a little confusing to me.  I suggest reformatting.

Because the title is missing a key word, I suggest looking over the paper once more for grammar and wording.

Author Response

Thank you for valuable comments concerning the manuscript. We provided the additional information and made appropriate corrections along the text.

Response to Reviewer 3 Comments

Comments and Suggestions for Authors

Title is missing the word, workers or staff.

Response: Thanks for the note, corrected now

The study examine the ability of Sputnik vaccines to protect medical staff from Covid-19.  Staff with 1,2 3 or four doses were surveyed with benefits found at all dose levels, although the small number for 3 to 4 doses is perhaps too small for any real conclusion.

Table 1 was a little confusing to me.  I suggest reformatting.

Response: Thank you for your comment! We corrected Table 1, removed the overall efficacy from the row of unvaccinated volunteers, and added a new row with the total number of volunteers vaccinated at least once during the follow-up, also we deleted excluded records from the table. According to the commets of other reviewers we changed data in this table to be more reflectable Delta variant dominance.

Because the title is missing a key word, I suggest looking over the paper once more for grammar and wording.

Response: We agree with this comment, we checked the text and corrected the style, corrected some terms and improved our English

Round 2

Reviewer 1 Report

The authors replied to my concerns and improved the manuscript.